# Decrease in Mobility during the COVID-19 Pandemic and Its Association with Increase in Depression among Older Adults: A Longitudinal Remote Mobility Monitoring Using a Wearable Sensor

**DOI:** 10.3390/s21093090

**Published:** 2021-04-29

**Authors:** Ramkinker Mishra, Catherine Park, Michele K. York, Mark E. Kunik, Shu-Fen Wung, Aanand D. Naik, Bijan Najafi

**Affiliations:** 1Interdisciplinary Consortium on Advanced Motion Performance (iCAMP), Michael E. DeBakey Department of Surgery, Baylor College of Medicine, Houston, TX 77030, USA; ram.mishra@bcm.edu (R.M.); catherine.park@bcm.edu (C.P.); 2Department of Neurology, Baylor College of Medicine, Houston, TX 77030, USA; myork@bcm.edu; 3Parkinson’s Disease Research Education and Clinical Center, Michael E. DeBakey Veterans Affairs Medical Center, 2002 Holcombe Boulevard, Houston, TX 77030, USA; 4Department of Psychiatry and Behavioral Sciences, Baylor College of Medicine, Houston, TX 77030, USA; mark.kunik@bcm.edu; 5VA Health Services Research and Development Center for Innovations in Quality, Effectiveness and Safety, Houston, TX 77021, USA; anaik@bcm.edu; 6VA South Central Mental Illness Research, Education and Clinical Center (a Virtual Center), Houston, TX 77021, USA; 7College of Nursing, University of Arizona, Tucson, AZ 85721, USA; wung@arizona.edu; 8Department of Medicine, Baylor College of Medicine, Houston, TX 77030, USA

**Keywords:** wearable sensor, COVID-19, pandemic, mobility, depression, older adults, telemedicine, digital health, frailty, mental health

## Abstract

Background: Social isolation during COVID-19 may negatively impact older adults’ wellbeing. To assess its impact, we measured changes in physical activity and sleep among community-dwelling older adults, from pre-to post-pandemic declaration. Method: Physical activity and sleep in older adults (n = 10, age = 77.3 ± 1.9 years, female = 40%) were remotely assessed within 3-month pre-to 6-month post-pandemic declaration using a pendant-wearable system. Depression was assessed pre-and post-pandemic declaration using the Center for Epidemiologic Studies Depression scale and was compared with 48 h continuous physical activity monitoring data before and during pandemic. Results: Compared to pre-pandemic, post-pandemic time spent in standing declined by 32.7% (Cohen’s d = 0.78, *p* < 0.01), walking by 52.2% (d = 1.1, *p* < 0.01), step-counts by 55.1% (d = 1.0, *p* = 0.016), and postural transitions by 44.6% (d = 0.82, *p* = 0.017) with increase in sitting duration by 20.5% (d = 0.5, *p* = 0.049). Depression symptoms increased by 150% (d = 0.8, *p* = 0.046). Interestingly, increase in depression was significantly correlated with unbroken-prolong sitting bout (ρ = 0.677, *p* = 0.032), cadence (ρ = −0.70, *p* = 0.024), and sleep duration (ρ = −0.72, *p* = 0.019). Conclusion: This is one of the early longitudinal studies highlighting adverse effect of the pandemic on objectively assessed physical activity and sleep in older adults. Our observations showed need for timely intervention to mitigate hard to reverse consequences of decreased physical activity such as depression.

## 1. Introduction

The coronavirus disease 2019 (COVID-19) pandemic has exacted great illness, death, distress, and unprecedented restrictions on our society [1]. Older adults are at high risk for developing serious complications from COVID-19 and have been recommended to follow strict guidelines to minimize the risk of exposure during the current pandemic. The resulting social isolation can have a significant negative impact on the physical and mental health of older adults [2,3], the extent of which is not completely known at this time. Understanding these effects can help mitigate the negative effects of social isolation caused by the pandemic and prevent long-lasting negative impacts on older adults’ health and wellbeing post-pandemic.

Previous studies suggest that, when older adults participate in social activities, it allows them to be physically active and preserve functional status [4,5]. However, as a preventive measure during the COVID-19 pandemic, community activities were halted, and social or family gatherings were discouraged. These measures may reduce physical activity level and sleep duration and, if they persist for a long time, could lead to hard to reverse consequences such as depression and frailty [6,7,8]. According to a report by the American Association of Retired Persons (AARP) and United Health Foundation, more than half of adults 50 years of age and older experienced social isolation during the pandemic, 41% felt more anxious, and 37% endorsed increased depressive symptoms [9]. A decline in mental health is often related to reduced physical activity, which increases the risk of mortality in older adults [3].

Pandemic-related restrictions can instigate a vicious cycle of reduced physical activity, abnormal sleep patterns, and decline in mental health in older adults [6], as shown in Figure 1. While systematic studies are lacking, Fitbit data scientists published the earliest report showing a 12% decline in the step count of millions of Fitbit users during the week of 22 March 2020, compared to the same week from the previous year [10]. Later, Browne et al. [11] demonstrated an 18% decline in daily steps count in 35 older adults with hypertension assessed before (January to March 2020) and during (June 2020) the COVID-19 pandemic. While Browne et al. demonstrated reduced physical activity in vulnerable older adults during the pandemic, they did not investigate the impact of COVID-19 on the sleep and mental health of older adults [11]. In one of the earliest surveys, AARP highlighted the impact of COVID-19 on the mental health of the general population and showed 41% of people were more anxious than usual, and 37% endorsed increased depressive symptoms [9]. Another survey conducted between 14 and 19 July 2020 by the Kaiser family foundation (KFF) showed that among, 1313 adults living in the United States, 36% of participants faced difficulty sleeping [12]. Furthermore, on analyzing electronic health records, it was observed that among 62,354 COVID survivors (including people with and without a past psychiatric diagnosis), 18% were diagnosed with higher anxiety or mood disorders within 14 to 90 days following the diagnosis of COVID-19 [13]. Altogether, COVID-19 related complications raise serious concern about the safety of older adults, and deterioration of mental and physical health can increase adverse events like falls. For instance, SafelyYOU’s community data indicate a 20% increase in falls for residents in memory care communities during the pandemic [14].

The above studies suggest that, during the pandemic, physical activity and sleep quality have deteriorated, probably because of voluntary (social isolation) and/or involuntary (lockdown) social distancing imposed by the pandemic. This deterioration may explain the higher prevalence of depression symptoms observed in older adults during the pandemic. However, the link between the increase in depression and changes in physical activity and sleep is still poorly understood. To address the gap, we leveraged an ongoing study initiated prior to the pandemic. More specifically, we recruited a cohort of community-dwelling older adults in which the patterns of physical activities and sleep were remotely monitored using a pendant sensor. When the pandemic was declared on 11 March 2020, we continued to follow-up with these participants up to 6 months post-pandemic, allowing us to objectively determine the pandemic’s effect on physical activities and sleep pattern. We analyzed the collected data to address two specific aims. In the first aim, we examined changes from pre- to post-pandemic in mobility performance, including commutative postures (sitting, standing, lying, and walking); walking characteristics (e.g., daily step count); motor performance (e.g., number of daily postural transition); and sleep (e.g., total sleep duration during night time). In the second aim, we explored the association between changes in mobility performance and depression during the same time interval. We hypothesized that there would be a decline in the participants’ physical activity and sleep with increased depression symptoms after the pandemic compared to pre-pandemic. Our second hypothesis was that increase in depression would be associated with reduced physical activity and sleep.

## 2. Materials and Methods

### 2.1. Study Population

Participants were recruited from an ongoing study focused on automatic fall detection and monitoring the risk of falling in community older adults with a high risk of falling using a wearable pendant sensor (ActivePERS/PAMSys, Biosensics LLC, Newton, MA, USA). Inclusion criteria were community older adults aged 75 years or older or aged 65 years older with a high risk of falling. The risk of falling was determined by either self-reported history of fall over last 12 months or self-report high concerns for fall. Participants were excluded if they were living in a nursing home, in hospice care, or if they were unable to independently walk a distance of 10 m with or without an assistive device or unable to stand still without moving feet, which may affect their daily physical activity, or were unwilling to participate. All participants were followed up for 12 months, including three time-point assessments of mobility, cognition, and mental health (e.g., depression) at baseline, 6 months, and 12 months. For this specific study, we excluded those without valid physical activity data or depression assessment no longer than three months before the pandemic declaration (11 March 2020) and with minimum three months to up to nine months post-pandemic declaration. The study was conducted according to the guidelines of the Declaration of Helsinki and approved by the Institutional Review Board (or Ethics Committee) of Baylor College of Medicine (protocol code H41717 and 13 December 2017).

### 2.2. Demographics and Clinical Data

Demographics and relevant clinical information, including age, gender, height, weight, ethnicity, fall history, and pre-existing medical conditions, were collected at baseline using chart-review and self-report. Body mass index (BMI) was calculated based on each older adult’s height and weight. At each study visit, participants underwent clinical assessments, including Montreal Cognitive Assessment (MoCA) [15], Center for Epidemiologic Studies Depression scale (CES-D) [16], Fear of falling (FES-I) [17], Lawton Instrumental Activities of Daily Living Scale (IADL) [18], and Beck Anxiety Inventory (BAI) [19]. The CES-D short-version scale was used to measure self-reported depression symptoms before and during the pandemic. A cutoff CES-D score of 16 or greater was used to identify subjects with depression [16]. Furthermore, scores of 36 and above on BAI indicated potentially concerning levels of anxiety in the participants [19]. Cognitive impairment was defined as a MoCA score less than 25, recommended by Nasreddine et al. (2005). Moreover, participants’ scores in the range of 28–64 on FES-I were considered to have a high concern about falling [20]. A person’s ability to perform tasks was determined based on a summary score on the IADL survey in the ranges from 0 (low function, dependent) to 8 (high function, independent) for women and 0 through 5 for men [18].

### 2.3. Sensor-Derived Monitoring of Physical Activity and Sleep

To collect fine grain-information about physical activity, we monitored physical activities and sleep pattern for a continuous period of 48h, at each study visit, using a validated pendant sensor (PAMSys™, BioSensics LLC, Watertown, MA, USA), worn around the neck (Figure 2). Physical activity and sleep in older adults were remotely assessed within the 3-month pre-to 6-month post-pandemic declaration during regular life. The participants were instructed to continuously wear the pendant for two consecutive days (48 h). Later, the pendant was returned to the research center through either a paid mail service or retrieved by a research coordinator. Prior studies have shown that two-day monitoring of daily life activities is sufficient to assess frailty [21] and yields optimal adherence to wearing the sensor continuously [22]. The PAMSys^TM^ pendant consists of a 3-axis accelerometer, battery, processor, and built-in memory for recording long-term data. Accelerometer signals were recorded at a sampling frequency of 50 Hz. All physical activity and sleep-related parameters were extracted from the pendant using two different validated algorithms; an algorithm to extract spontaneous daily physical activities (e.g., cumulative postures, postural transitions, and walking characteristics) and an algorithm to quantify sleep quality (e.g., sleep/awake duration during night-time). These algorithms were described in detail in our previous studies [23,24,25,26].

The estimated physical activities included for final data analysis were characterized by (1) cumulated percentage of sitting, standing, lying, and walking postures; (2) daily walking characteristics (step count, cadence), and the number of the unbroken walking bouts, which included a minimum three consecutive steps within a 5-s interval [27]); (3) number of postural-transitions summing up postural-transitions such as sit-to-stand, stand-to-sit, walk-to-stand, stand-to-walk, walk-to-sit. When the duration of standing between a walk to sit and vice versa was less than 1 s, it was considered as walk-to-sit or sit-to-walk transition [28]; and (4) activity behavior, including unbroken prolonged sitting bout, light activity, and moderate-to-vigorous activity [21,25]. The unbroken prolonged sitting bout was defined as the 90th percentile of all recorded sitting bouts duration [21,25]. Light activity was defined as an activity between metabolic equivalent (MET) ≥1.5 and <3.0, such as ironing, washing, and dusting, and working at a standing workstation [19]. Moderate-to-vigorous activity referred to an activity demanding ≥3.0 MET, such as brisk walking, recreational activities, taking stairs, etc. [21].

Furthermore, sleep was characterized in terms of the total duration of a participant’s time in bed during the night [21]. We described the details for extracting time in bed during the night in our previous study [26]. Briefly, first, the unwanted noise was removed from the acceleration signal by applying a band-pass filter. Then, for every minute, a vector magnitude/norm of acceleration was estimated. Finally, a model was used to estimate the sleep/wake conditions based on the moment and standard deviation calculated from every one-minute acceleration vector, posture (sleeping on sides or back), and postural transition (e.g., tossing on bed, rotating from back to sides, etc.) information.

### 2.4. Statistical Analysis

All continuous data were presented as the mean ± standard error, and categorical data were expressed as the percentage. A paired *t*-test was used for with-group comparison of continuous demographics, clinical data, and physical activity metrics. Before applying the paired *t*-test, the assumption of normality was assessed using Shapiro–Wilk’s test of normality (*p* > 0.05). The Wilcoxon signed-rank test was performed if the normality assumption was not satisfied. The effect size for discriminating between groups was estimated using Cohen’s d effect size and represented as d [29]. The Spearman correlation coefficient was used to evaluate the degree of association between physical activity metrics and depression level. The correlation coefficient was also interpreted as effect size [30]. All statistical analyses were performed using IBM SPSS Statistics 25 (IBM, Chicago, IL, USA), with a significance level defined as *p* < 0.05.

## 3. Results

### 3.1. Demographic and Clinical Characteristics

Ten older adults (age = 77.3 ± 1.9 years, females = 40%, and BMI = 27.5 ± 1.6 kg/m^2^) satisfied the inclusion and exclusion criteria of this study. On average, the baseline data were collected 1.13 ± 0.43 months before the pandemic, and follow-up assessment performed on average 5.9 ± 0.67 months post-pandemic declaration. Table 1 summarizes the demographics and clinical information of these participants. At baseline, participants reported low levels of depression (2.8 ± 0.7) and anxiety (2.1 ± 1.0) symptoms on CES-D and BAI scales, respectively. According to the MoCA assessment, three participants (30%) were classified as cognitively impaired. Furthermore, two participants (20%) reported having a low functional ability on IADL. While three participants (30%) reported falls in the last 12 months, FES-I scores indicated no participant had a high concern of falling. As presented in Table 2, post-pandemic declaration, there was an increase in depression symptoms by 150% (Cohen’s d = 0.8, *p* = 0.046), but no significant change in other psychosocial metrics (e.g., anxiety, fear of falling, and activities of daily life).

### 3.2. Physical Activity and Sleep Characteristics

There was an increase in daily sitting duration by 20.5% (d = 0.5, *p* = 0.049). Furthermore, there was a decline in daily standing duration on average by 32.7% (d = 0.78, *p* < 0.01), walking duration by 52.2% (d = 1.1, *p* < 0.01), daily step-counts by 55.1% (d = 1.0, *p* = 0.016), and number of daily postural transitions by 44.6% (d = 0.82, *p* = 0.017) (Table 2, Figure 3 and Figure 4). There was no significant change in other sensor-derived parameters.

### 3.3. Association between Change in Depression and Sensor-Derived Parameters

Table 3 summarizes the correlations between change in depression score with change in cadence, unbroken prolonged sitting bout, activity behavior, and sleep. As represented in Figure 5, we observed that as depression symptoms increased, there was a decline in cadence (ρ = −0.701, *p* = 0.024), time in bed (ρ = −0.72, *p* = 0.019) with a trend towards reduced light activity (ρ = −0.566, *p* = 0.088), and moderate-to-vigorous activity (ρ = −0.41, *p* = 0.24) levels. Furthermore, increase in depression score was associated with increase in prolonged sitting (ρ = 0.68, *p* = 0.032).

## 4. Discussion

This study explored longitudinal objective changes in patterns of physical activities and sleep from pre- to post-pandemic in community-dwelling older adults and associate them with the increase in depression symptoms as summarized in Figure 6. The major findings of the present study were a significant decline in physical activity pattern including cumulative postures (sitting, standing, and walking), walking quantity (daily step count), high-level motor function (daily number of postural transitions), and sleep quantity (time in bed during night). Furthermore, it revealed a noticeable pairwise increase in depression score on average by 150% within 6 months post-pandemic compared to 3 months pre-pandemic. Most importantly, this study found a significant association between the increase in depression symptoms post-pandemic with deterioration in physical activity and sleep. For instance, the results suggest that 52%, 49%, 46%, and 32% of the variation in depression score from pre- to post-pandemic could be explained by reduced time in bed at night, a decline in cadence, increase in the unbroken prolonged sitting bout, and reduction in the percentage of light activities from pre- to post-pandemic declaration. These findings are aligned with the conclusions of Sepúlveda-Loyola et al. (2020), who suggest reduced social interaction due to COVID-19 related restrictions increases depression and reduces physical activity level in older adults [5]. These results confirm the expected unhealthy physical activity level changes and decline in older adults’ mental health during the COVID-19 pandemic.

Several studies based on the surveys, interviews, and questionnaires support the negative impact of COVID-19 on physical activity patterns, sleep, and psycho-social behavior in older adults [5,6,9,11]. To our knowledge, this is the first prospective study that reports objective sensor-derived data related to physical activities and sleep patterns from the same individual followed from the pre-declaration of the pandemic to several months post COVID-19 pandemic declaration. This study design enables untangling of the association between the changes in physical activity and sleep patterns to changes in depression because of the COVID-19 pandemic in older adults. Our study suggests that increased depression during the COVID-19 pandemic in older adults can be explained by increased prolonged unbroken sitting bout, reduction in cadence during walking, and reduction in sleep duration during the night. Our results also suggest that during the COVID-19 pandemic, the daily step count was reduced on average by 55.1% during the pandemic compared to pre-pandemic, probably because of voluntary or involuntary social restriction imposed by the pandemic. During the same time, the depression symptom was increased by 150%. Together, we speculate that the increase in depression was partially because of a reduction in the level of physical activities. This speculation is supported by few pre-pandemic prospective studies in which it was demonstrated that a decline in the level of physical activities might lead to an increase in depression symptoms. For instance, Edwards and Loprinzi [31] recruited 26 young adults (age: 21.7 ± 2.71 years, 38% male). The participants were asked to keep their walking steps count to less than 5000 steps per day for a one-week duration. The results suggest that this restriction led to an increase in depression symptoms on average by 83.7% (*p* < 0.01). In another prospective study, Endrighi et al. [32] recruited 43 younger adults between 18 and 35. During the first two weeks, participants were asked to be as sedentary as much as possible. Then, for the follow-up two weeks, they were asked to return to their daily routine. Using ActiGraph, they showed that sedentary time was on an average 5.5% higher during the first two weeks compared to the habitual activities (during the next two weeks). The results suggest that this increase in sedentary time led to a 55.1% increase in psychological distress and mood disturbance [32].

Even though a progressive return to everyday life is taking place gradually, all the preventive and safety measures for the ongoing pandemic are instilling feelings of uncertainty, fears, and concerns. Furthermore, limited available resources for older adults to mitigate mental health concerns pose a more significant challenge in the current circumstances. As implied by our results, increased sedentary behavior in older adults should be a target for treatment management during the COVID-19 pandemic. Prolonged reduction in physical activity and sleep could lead to hard to reverse complications such as depression, cognitive decline, and frailty [3,19].

Our previous study showed that people with an increased level of frailty (non-frail, pre-frail to frail state) exhibit a reduced number of postural transitions in daily life activities [26]. In the context of COVID-related changes, the postural transition can be considered a more reliable measurement of functional performance in older adults less influenced by environmental conditions compared to daily step counts or walking duration [33].

This study has limitations that need to be acknowledged. First, although we observed pairwise changes in depression level and physical activity level with medium to large effect sizes, the sample size was small; therefore, our results should be interpreted with caution. Second, this is a secondary analysis of an ongoing study, which was designed to remotely monitor the physical activity and fall incidents with a timeline of 12 months; thus, it may be underpowered to validate this study’s hypotheses. Participants screened for this study, with their assessment time points overlapping with before and during the pandemic, were used in this exploratory analysis. Therefore, the duration between pre-and post-pandemic assessments varied from 4 to 12 months (Table 1). Another limitation is that, for given study design, the directionality between change in depression and physical activity, i.e., it may be that depression is causing decline in physical activity. Despite the mentioned limitations above, our preliminary findings might contribute to a better understanding of the unhealthy physical and mental health changes during the COVID-19 pandemic and guide future feasible preventive and therapeutic actions for older adults. In the future, a systematic study would be needed that can investigate changes in physical and mental health in older adults during and after the pandemic.

Despite the above limitations, this study may better shed light on the effect of the COVID-19 pandemic on physical activity, sleep, and depression thanks to its longitudinal study design. Another key advantage of our study is the use of objective sensor-derived metrics design enabling us to objectively track for the same individual the changes in physical activities and sleep patterns from the pre-pandemic to few months post the declaration of pandemic and link it to changes in depression symptoms.

## 5. Conclusions

This prospective study highlighting the adverse effect of the COVID-19 pandemic on physical activity and sleep, objectively monitoring using a pedant sensor in community dwelling older adults. We observed a 150% increase in depression, and this increase is correlated with the prolonged sitting bout, nighttime sleep duration, and cadence. Additionally, our results suggest that reduced sleep time explains the 52% variance in change in depression. Timely intervention strategies are urgent to support recovery in physical activity and functional performance to the pre-pandemic level to mitigate the consequences of social isolation, including the increase in depression. The critical limitation includes the small sample size and timeline of assessment. In the future, a systematic study would be needed that can investigate changes in physical and mental health in older adults during and after the pandemic.

## Figures and Tables

**Figure 1 sensors-21-03090-f001:**
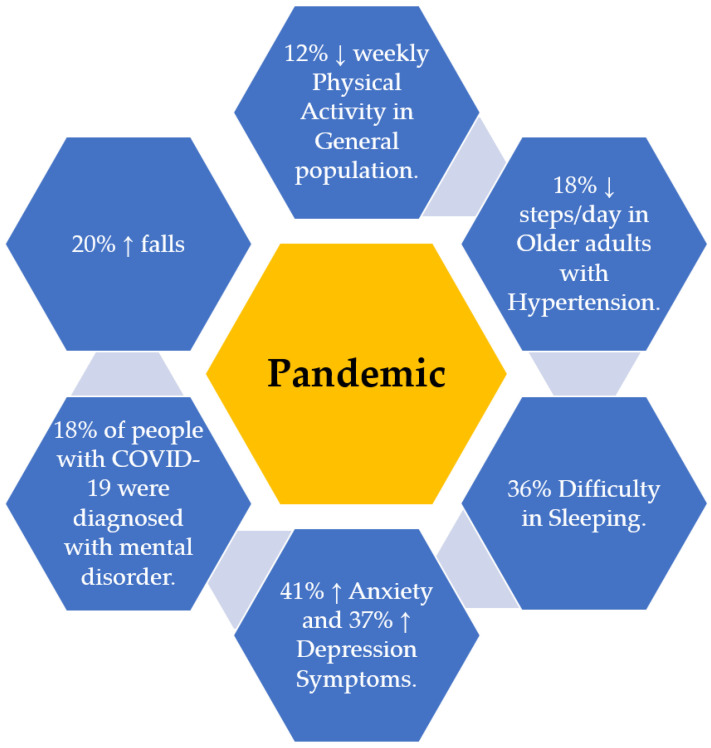
Summarized representation of studies which investigated the impact of the COVID-19 pandemic on physical activity, sleep, and mental health. These studies highlighted that, during the COVID-19 pandemic, the activity level has been reduced in both the general population [10] and those with chronic illness (e.g., hypertension) [11], and sleep quality has deteriorated [12]. Other studies suggested that during the pandemic, depression, anxiety, and mental health disorder have been increased compared to pre-pandemic [9,13,14]. However, these studies were limited to create a link between changes in the pattern of physical activities and sleep with changes in depression symptoms in the context of a prospective study design.

**Figure 2 sensors-21-03090-f002:**
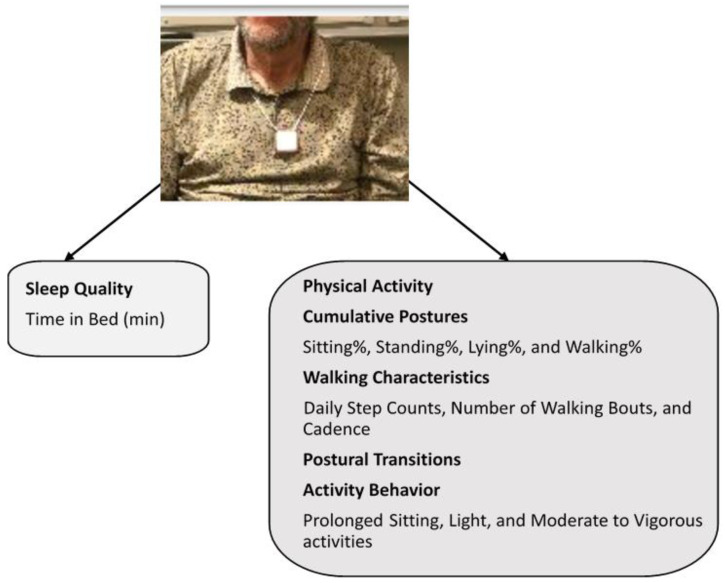
Participants wore the pendant sensor around the neck as shown above and the physical activity and sleep parameters that were extracted based on the validated algorithm are shown below, namely: time in bed, cumulative postures, walking characteristics, postural transitions, and activity behavior.

**Figure 3 sensors-21-03090-f003:**
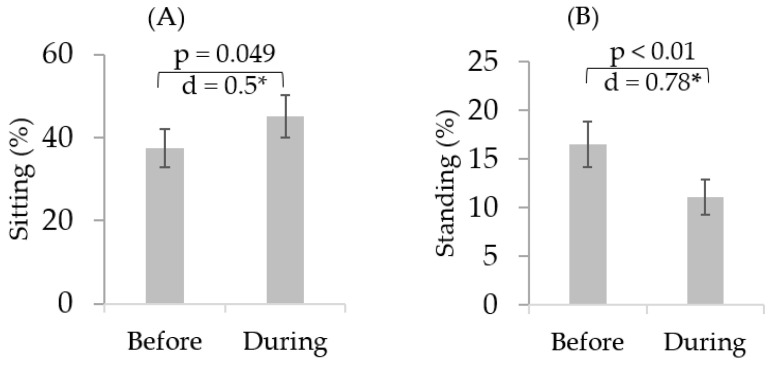
Representation of percentage of cumulated time spent in postures involving (**A**) sitting; (**B**) standing; (**C**) walking; and (**D**) Lying by the participants as remotely assessed by the PAMSys sensor before and during the pandemic. While there was no significant change in overall time spent lying, there was a significant increase in sedentary behavior and a decline in time spent walking or standing. * represents significant difference (*p* < 0.05).

**Figure 4 sensors-21-03090-f004:**
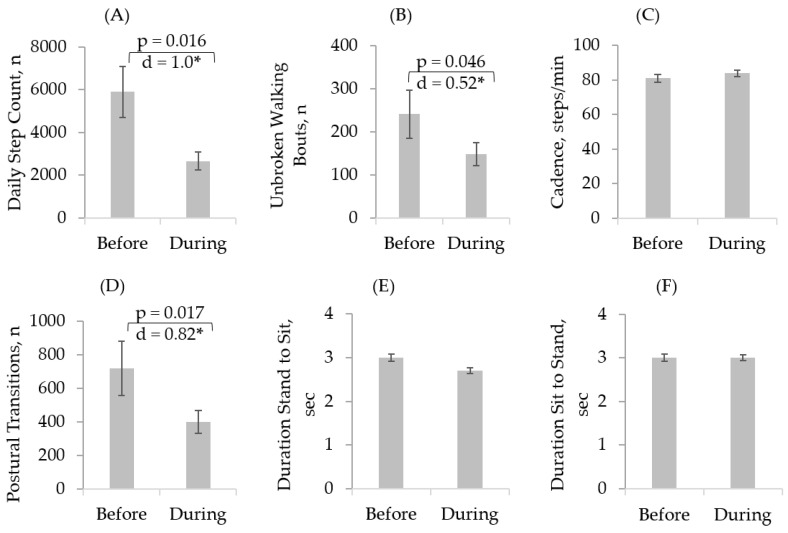
Walking characteristics and postural transitions of the participants as remotely assessed by the PamSys sensor before and during pandemic. We observed a significant decrease in (**A**) daily step count, (**B**) unbroken walking bout, and (**D**) postural transitions. There was no significant change on (**C**) cadence, (**E**) duration of stand to sit transition, and (**F**) duration of sit to stand transition. Error bars indicate standard errors of the corresponding averages (* represents significant differences (*p* < 0.0.05)).

**Figure 5 sensors-21-03090-f005:**
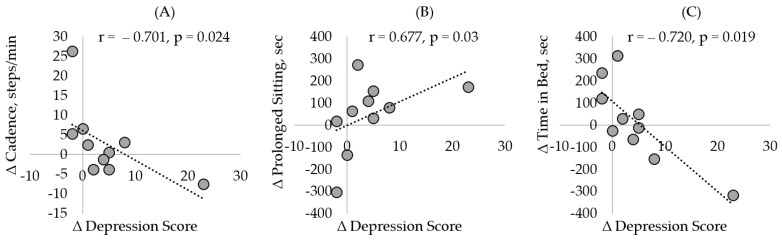
Association between change in depression and sensor derived parameters. Increase in depression symptoms was significantly correlated with (**A**) change in cadence, (**B**) change in prolonged sitting, and (**C**) change in time in bed. The symbol ∆ represents the change in metric (During‒Before pandemic).

**Figure 6 sensors-21-03090-f006:**
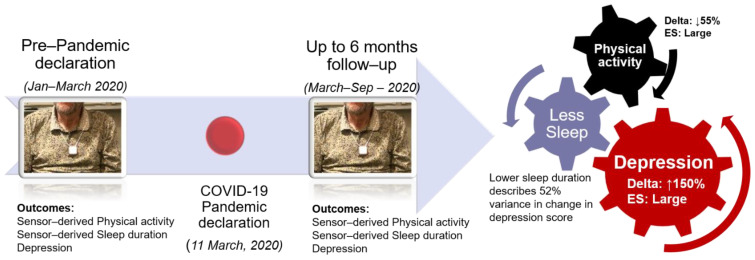
In this study, we followed a cohort of community-dwelling older adults (n = 10) from the pre-pandemic to up to 6 months post declaration of the COVID-19 pandemic (11 March 2020). The results suggest that, compared to pre-pandemic physical activity (daily step count) it is reduced by 55%, and depression is increased on average by 150%. Our results also suggest that lower sleep duration (time in bed during the night-time) describes 52% of the variance in change in depression score. Abbreviation: ES represents effect size.

**Table 1 sensors-21-03090-t001:** Demographic and clinical information of the participants. Values are presented as mean ± standard error (SE), or percentage (%).

Demographics	
Age, years	77.3 ± 1.9
Sex (Female), %	40%
Height, m	1.63 ± 0.09
Weight, kg	83.4 ± 21.5
Body Mass Index (BMI), kg/m^2^	27.5 ± 1.6
**Clinical data**	
Had fall in last 12-month, %	30%
Cancer (%)	40%
Number of prescription medications, n	2.1 ± 0.6
Cognition (MoCA), score	25.1 ± 1.6
Cognitive impairment, %	30%
Center for Epidemiologic Studies Depression (CES-D), score	2.8 ± 0.7
Depression, %	0%
Fear of Falling (FES-I), score	19.3 ± 1.0
(High Concern) Fallers, %	0%
Activities of Daily Living Scale (IADL), score	6.6 ± 0.9
Low Functional Ability, %	20%
Anxiety (BAI), score	2.1 ± 1.0
High Anxiety, %	0%
**Time-point of assessments**	
Average duration pre-pandemic assessment, months	1.13 ± 0.43
Average duration post-pandemic assessment, months	5.9 ± 0.67

**Table 2 sensors-21-03090-t002:** Psycho-social, Cumulated Posture, Walking characteristics, Postural transitions, Activity behavior, and Sleep related measures assessed before and during pandemic.

	Before	During	Mean Difference %	Cohen’s d	*p*-Value
**Psycho-social Behavior**					
Depression, score	3.0 ± 0.7	7.5 ± 2.4	150.0%	0.80	0.046 *
Fear of Falling, score	19.7 ± 1.2	18.7 ± 1.0	−5.1%	0.29	0.443
Anxiety, score	2.1 ± 1.0	2.9 ± 1.4	38.1%	0.21	0.588
Activity of daily life, score	6.6 ± 0.9	6.0 ± 1.0	−9.1%	0.19	0.131
**Cumulated Posture**					
Sitting percentage, %	37.5 ± 4.5	45.2 ± 5.1	20.5%	0.5	0.049 *
Lying percentage, %	39.3 ± 4.6	40.5 ± 4.7	3.1%	0.24	0.768
Standing percentage, %	16.5 ± 2.3	11.1 ± 1.8	−32.7%	0.78	<0.01 *
Walking percentage, %	6.7 ± 1.3	3.2 ± 0.5	−52.2%	1.1	<0.01 *
**Walking Characteristics**					
Daily Step count, n	5911 ± 1193	2655 ± 419	−55.1%	1.0	0.016 *
Number of unbroken walking bout, n	241.3 ± 56.2	148.6 ± 27.1	−38.4%	0.52	0.046 *
Cadence, steps/min	81.1 ± 2.3	83.9 ± 1.7	3.4%	0.4	0.367
**Postural Transition**					
Number of Postural Transitions, n	720.7 ± 162.2	399.4 ± 68.5	−44.6%	0.82	0.017 *
Average duration of stand-to-sit transition, s	3.0 ± 0.07	2.7 ± 0.3	−10.0%	0.4	0.88
Average duration of sit-to-stand transition, s	3.0 ± 0.08	3.0 ± 0.07	0%	0	0.57
**Activity Behavior**					
Prolong Sitting, s	240.9 ± 46.5	287.3 ± 61.6	19.3%	0.26	0.392
Average Light Activity, min	10.8 ± 0.7	10.5 ± 0.8	−2.8%	0.13	0.678
Average Moderate to Vigorous Activity, min	31.0 ± 5.4	27.3 ± 4.6	−11.9%	0.23	0.526
**Sleep Quantity**					
Time in Bed, s	566.3 ± 66.2	583.7 ± 67.5	3%	0.08	0.768

* represents the significant difference (*p* < 0.05).

**Table 3 sensors-21-03090-t003:** Association with the change in depression measured using Spearman correlation analysis.

	Correlations Coefficient	Variance, R^2^	*p*-Value
∆ ^1^ Cadence, steps/min	−0.701 *	0.49	0.024
∆ Prolonged Sitting, s	0.677 *	0.46	0.032
∆ Average Light Activity, min	−0.566	0.32	0.088
∆ Average Moderate to Vigorous Activity, min	−0.409	0.16	0.241
∆ Time in Bed, s	−0.720 *	0.52	0.019

^1^ ∆ represents change in metric (During–Before pandemic). * represents the significant (*p* < 0.05) correlation with the ∆ Depression.

## Data Availability

The data presented in this study are available on request from the corresponding author. The data are not publicly available due to propriety reasons and need for IRB permission before sharing.

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
