# Peer review of "Decrease in Mobility during the COVID-19 Pandemic and Its Association with Increase in Depression among Older Adults: A Longitudinal Remote Mobility Monitoring Using a Wearable Sensor"

_sensors, 2021, doi:10.3390/s21093090_

Round 1

Reviewer 1 Report

Paper deal with an interesting problem where experimental investigation is presented, evaluated.

Paper has good potential, but current presentation in still not in the level for this journal yet. SO please update at minimum:

  • results are presented in clear way, but not compared to any other studies..
  • results need to be in context with other studies.. papers..
  • You need to extend section Discussion, where you will sumarize contribution and findings..  to show clearly comparison.. pros and cons.. and to discuss also other studies

Conclusions need to contain PROS and CONS of your solution using also VALUES.. Contribution.. and future direction..

references need to contain more journal sources from Q1/Q2.. 

Author Response

Response to Reviewer 1 Comments

Point 1: Paper deal with an interesting problem where experimental investigation is presented, evaluated.

Paper has good potential, but current presentation in still not in the level for this journal yet. SO please update at minimum: 

 Response 1: We appreciate reviewer’s feedback. The revision has highly benefited from the reviewer comments and suggestions as described below.

  • To improve introduction section, we included figure 1 representing summarized representation of studies which investigated the impact of COVID-19 pandemic on physical activity, sleep, and mental health.
  • Furthermore, we modified the text to provide further details for the studies done so far to investigate the impact of COVID-19 on page 1, line number 75-103.
  • To better present the results, we included figure 3, 4, and 5.
  • To better represent method, we included figure 6 in the discussion section that present methodology and main findings of the present study in the concise manner.

Point 2: results are presented in clear way, but not compared to any other studies.. results need to be in context with other studies.. papers..

Response 2: We appreciate the thoughtful suggestion. To our knowledge there is no prospective study that objectively track changes in the level of physical activity as well as depression symptoms for the same individual from the pre-pandemic to few months post the declaration of the pandemic. However, we identified few relevant prospective studies [32, 33] pre the pandemic that support our conclusions. These studies were included in the revised discussion section. Additionally, figure3, 4, and 5 are included to present the results in the better way.

Following paragraph on page 9-10 Line number 267-295 is added:

“Several studies based on the survey, interview, and questionnaires support the negative impact of COVID-19 on physical activity patterns, sleep, and psycho-social behavior in older adults [5,6,9,11,15]. To our knowledge, this is the first prospective study that reports objective sensor-derived data related to physical activities and sleep patterns from the same individual followed from the pre-declaration of the pandemic to several months post COVID-19 pandemic declaration. This study design enables to untangle the association between the changes in physical activity and sleep patterns to changes in depression because of the COVID-19 pandemic in older adults. Our study suggests that increased depression during the COVID-19 pandemic in older adults can be explained by increased prolonged unbroken sitting bout, reduction in cadence dur-ing walking, and reduction in sleep duration during the night. Our results also suggest that during the COVID-19 pandemic, the daily step count was reduced on average by 55.1% during the pandemic compared to pre-pandemic, probably because of voluntary or involuntary social restriction imposed by the pandemic. During the same time, the depression symptom was increased by 150%. Together we speculate that the increase in depression was partially because of a reduction in the level of physical activities. This speculation is supported by few pre-pandemic prospective studies in which it was demonstrated that a decline in the level of physical activities might lead to an increase in depression symptoms. For instance, Edwards and Loprinzi [32] recruited 26 young adults (age: 21.7 ± 2.71 years, 38% male). The participants were asked to keep their walking steps count to less than 5000 steps per day for a one-week duration. The re-sults suggest that this restriction led to an increase in depression symptoms on average by 83.7% (p<0.01). In another prospective study, Endrighi et al. [33] recruited 43 younger adults between 18 to 35 years older adults. During the first two weeks, they asked participants to be as sedentary as much as possible. Then for the follow-up two weeks, they were asked to return to their daily routine. Using ActiGraph, they showed that sedentary time was on an average 5.5% higher during the first two weeks com-pared to the habitual activities (during the next two weeks). The results suggest that this increase in sedentary time led to 55.1% in psychological distress and mood dis-turbance [33].

Point 3: You need to extend section Discussion, where you will sumarize contribution and findings..  to show clearly comparison.. pros and cons.. and to discuss also other studies.

Response 3: Please see our response to Point 2.

Point 4: Conclusions need to contain PROS and CONS of your solution using also VALUES.. Contribution.. and future direction.

Response 4: The conclusion is modified as suggested. The primary findings, limitations, advantages, and future direction of present findings are provided in the conclusion section. Additionally, text about pros of this study is included in the discussion section on page 11, line number 326-331. We also included figure 6 which provide summarized information about methodology and main findings of the present study.

Conclusion Section (line number 333-343):

This prospective study highlighting the adverse effect of the COVID-19 pandemic on physical activity and sleep, objectively monitoring using a pedant sensor in com-munity dwelling older adults. We observed a 150% increase in depression, and this in-crease is correlated with the prolonged sitting bout, nighttime sleep duration, and ca-dence.  Additionally, our results suggest that reduced sleep time explains the 52% variance in change in depression. Timely intervention strategies are urgent to support recovery in physical activity and functional performance to the pre-pandemic level to mitigate the consequences of social isolation, including the increase in depression. The critical limitation includes the small sample size and timeline of assessment. In the fu-ture, a systematic study would be needed that can investigate changes in physical and mental health in older adults during and after the pandemic.”

Point 5: references need to contain more journal sources from Q1/Q2..

Response 4: We included the most relevant citations. We recognize that because of limited studies exploring the effect of the pandemic and the relatively small sample size of these studies, probably because of the challenge of conducting clinical research during the pandemic, very few relevant studies [32, 33] are currently available in the literature. If, however, we are welcoming and appreciate any missing studies that may be relevant to the scope of this study.

Reviewer 2 Report

Authors study the influence of the pandemic situation on mobility and depression in adults.

The article is well written and structured. 

In-vivo experiments have been conducted and statistical analysis has been applied on the collected data. 

The article is technically sound. Statistical data analysis has been correctly applied, and the obtained results  have been extensively discussed in paragraph 4.

My only concern regards the number of day each experimental subject has worn the sensor. Since only two days have been considered for the experiments, how did you evaluated the differences between pre/post pandemic period?

Did each subject wear two times the sensor, in order to obtain data in the different phases of the pandemic?

This must be better detailed in the manuscript, since it is not clear. 

Finally, will you plan to make your data available (after anonymization)? 

Author Response

Response to Reviewer 2 Comments

Point 1: Authors study the influence of the pandemic situation on mobility and depression in adults.

The article is well written and structured.

In-vivo experiments have been conducted and statistical analysis has been applied on the collected data.

The article is technically sound. Statistical data analysis has been correctly applied, and the obtained results  have been extensively discussed in paragraph 4.

Response 1: We appreciate reviewer’s feedback.

  • To improve introduction section, we included figure 1 representing summarized representation of studies which investigated the impact of COVID-19 pandemic on physical activity, sleep, and mental health.
  • Furthermore, we modified the text to provide further details for the studies done so far to investigate the impact of COVID-19 on page 1, line number 75-103.

Point 2: My only concern regards the number of day each experimental subject has worn the sensor. Since only two days have been considered for the experiments, how did you evaluated the differences between pre/post pandemic period?

Response 2: Participant wore the sensor two times pre- and post-pandemic. To avoid confusion, we included lines 158-159 as following:

Physical activity and sleep in older adults were remotely assessed within 3-month pre-to 6-month post-pandemic declaration, during regular life.”

Point 3: Did each subject wear two times the sensor, in order to obtain data in the different phases of the pandemic?

Response 2: Yes, data collection was done twice (pre- and post- pandemic declaration). Additionally, to better represent method and findings, we included figure 6 in the discussion section that present methodology and main findings of the present study in the concise manner.

Point 3: Finally, will you plan to make your data available (after anonymization)??..

Response 3: Yes, we can provide the data on request.

Reviewer 3 Report

The authors present an interesting study in which they examined both changes from pre- to post-pandemic in mobility performance and the association between changes in mobility performance and depression.

The design of the protocol and the data analysis are good. Although the study has some limitations, these were all mentioned properly in the discussions section.

The contents of the manuscript can draw a lot of attention from several research groups. Indeed, many similar studies aimed to understand all the harmful collateral aspects that this pandemic has created are currently ongoing.

Few minor revision are listed below:

MINOR REVISION

Please, check that there is not a double space in lines 51 and 79;

Please, correct “March 2021” with “March 2020” in line 77;

Please, consider a possible variation of punctuation in line 97;

Please, specify within the text if the subjects enrolled for the tests have provided authors with a written report on their activities during the 48 hours;

It would be useful a generic description of the dimensions of the rooms/environments/conditions to which the different subjects had access during the pandemic

Author Response

Response to Reviewer 3 Comments

Point 1: The authors present an interesting study in which they examined both changes from pre- to post-pandemic in mobility performance and the association between changes in mobility performance and depression.

The design of the protocol and the data analysis are good. Although the study has some limitations, these were all mentioned properly in the discussions section.

The contents of the manuscript can draw a lot of attention from several research groups. Indeed, many similar studies aimed to understand all the harmful collateral aspects that this pandemic has created are currently ongoing.

Response 1: We appreciate reviewer’s feedback. The revision has highly benefited from the reviewer comments and suggestions as described below.

  • To improve introduction section, we included figure 1 representing summarized representation of studies which investigated the impact of COVID-19 pandemic on physical activity, sleep, and mental health.
  • Furthermore, we modified the text to provide further details for the studies done so far to investigate the impact of COVID-19 on page 1, line number 75-103.
  • To better represent method, we included figure 6 in the discussion section that present methodology and main findings of the present study in the concise manner.

Point 2: Please, check that there is not a double space in lines 51 and 79;

Response 2: We followed the template provided by the sensor and it is not a double space..

Point 3: Please, consider a possible variation of punctuation in line 97;

Response 3: We replaced the punctuation “;” now line number 126 with commas “,”.

Point 4: Please, specify within the text if the subjects enrolled for the tests have provided authors with a written report on their activities during the 48 hours;

Response 4: Since this is an observational study with usual, to avoid any bias, we didn’t report about physical activities to participants. However, patients with severe depression were referred to our clinical collaborators for counselling according to our IRB protocol.

Point 5: It would be useful a generic description of the dimensions of the rooms/environments/conditions to which the different subjects had access during the pandemic.

Response 5: The objective physical activity and sleep data was collected using pendant sensor which was used by the participants in everyday life. The data collection was done for 48 hours pre- and post-pandemic declaration. To bring more clarity following line is added in “Physical activity and sleep in older adults were remotely assessed within 3-month pre-to 6-month post-pandemic declaration, during regular life.”

method (line number 158-159).

Additionally, figure 6 is included to provide concise detail about the study design and main findings of the study.